# Moving Morehouse School of Medicine Translation T^x^ Research through MDTTs—Multidisciplinary Translational Teams

**DOI:** 10.3390/ijerph20054302

**Published:** 2023-02-28

**Authors:** Rhonda Conerly Holliday, Kendra D. Piper, Shawn X. Trimble, Carmen M. Dickinson-Copeland, Ashley K. Mitchell, Tabia Henry Akintobi, Vincent C. Bond, Virginia D. Floyd

**Affiliations:** Morehouse School of Medicine, 720 Westview Dr SW, Atlanta, GA 30310, USA

**Keywords:** translational research, multidisciplinary teams, health equity, community participatory research

## Abstract

Morehouse School of Medicine (SOM) works to achieve its vision of advancing health equity through conducting transformational, translation science (T^x^). T^x^ describes our translational research continuum, symbolizing a method and scientific philosophy that intentionally promotes and supports convergence of interdisciplinary approaches and scientists to stimulate exponential advances for the health of diverse communities. Morehouse SOM actualizes T^x^ through multidisciplinary translational teams (MDTTs). We chronicle the identification of MDTTs by documenting formation, composition, functioning, successes, failures, and sustainability. Data and information were collected through key informant interviews, review of research documents, workshops, and community events. Our scan identified 16 teams that meet our Morehouse SOM definition of an MDTT. These team science workgroups cross basic science, clinical, and public health academic departments, and include community partners and student learners. We present four MDTTs, in various stages of progress, at Morehouse SOM and how they are advancing translational research.

## 1. Introduction

The vision of Morehouse School of Medicine (Morehouse SOM) is straightforward: To lead the creation and advancement of health equity. Our Morehouse SOM team science process utilizes an approach to increase the rapid translation of research discoveries to improve the health of minority, underserved populations and/or disadvantaged communities. Morehouse SOM works to achieve health equity through conducting T^x^ transformational translation science. T^x^ is the Morehouse SOM description of our translation research continuum. Coined at Morehouse SOM, T^x TM^ symbolizes a method and scientific philosophy that intentionally promotes and supports convergence of interdisciplinary approaches and scientists to stimulate exponential advances for the health of diverse communities [1]. The “T” of the trademark acknowledges the importance of the phases of the translational continuum. However, the exponent “x” represents the goal to move research from the Translational to the Transformational. Akintobi et al., 2019 describes in detail the approach and philosophy of T^x^ at Morehouse SOM in the advancement of health equity [1]. The T^x^ continuum at Morehouse SOM is realized through team science.

Team science is a well-researched paradigm to address complex health issues, health disparities and advance health equity [2,3,4]. It involves researchers from different disciplines working collaboratively to address health on the translational science continuum. In its earliest iteration it was seen as the combination of basic and clinical science researchers coming together to link research findings, breaking down the academic silos which usually define and constrict the full implementation (translation) of basic science research into meaningful clinical and community intervention practices. Team science has become more frequent as we address scientific challenges that are increasingly more complex coupled with complex societal challenges. These challenges require diverse expertise to continue to drive impactful scientific discovery, translation, and transformation, and has led to increased investment in this strategy [5].

Effective team science has, at its’ foundation, trust. Other characteristics of successful teams include a shared vision, strategic identification of members, comfort with disagreement, conflict management, and clear expectations regarding acknowledgement of contributions and authorship [6,7]. A review of team science literature has found that teams which are organizationally (both departmentally and institutionally) diverse, geographically diverse, and cross-disciplinary produce impactful research [8]. Impactful research, measured by publications, is produced by teams that are large and demonstrate diversity in gender, academic rank and profession, while less is known about the impact of ethnic and cultural diversity [8].

The nomenclature for these teams varies from institution to institution. At Morehouse SOM, the approach taken to team science is called Multidisciplinary Translational Teams (MDTTs). To accomplish the Morehouse SOM T^x^ research agenda, we realized that a new work process structure would be necessary. We have defined an MDTT as a core team of researchers that include members from the disciplines of both basic and clinical research and, unlike standard team science definitions, must include a student learner and community member, representing an expanded definition of team science. The inclusion of the student learner and community member is in line with Morehouse SOM mission and core values to train future scientists and healthcare providers, and to engage the community in meaningful ways as active participants and not passive recipients. It is our hypothesis that conducting T^x^ translation research through these types of expanded teams will facilitate the translation of basic research to community interventions more quickly and with wider dissemination, particularly for minority communities who are often saddled with a greater disease burden, ultimately advancing health equity.

Until recently, the systematic identification and promotion of MDTTs at Morehouse SOM had not been undertaken. Institutional factors, including the provision of resources, environmental structures to foster collaboration, and implementation of organizational structures are important in establishing and promoting team science [8]. The goals of this paper are to describe the institutional process undertaken to define, identify, construct, and encourage MDTTs at Morehouse SOM, to describe the importance of team science in reaching Morehouse SOM’s vision of achieving health equity, and to understand the role of expanded multidisciplinary team science in increasing the journey from basic science and public health research to patient therapies and community interventions.

## 2. Detailed Case Descriptions

Through U54 Research Centers at Minority Intuitions (RCMI) funding, Morehouse SOM established the Center for Translational Research in Health Disparities (CTRHD) and was supported to identify, encourage, and build MDTTs throughout the institution. A cross-departmental MDTT workgroup representing basic, clinical, and behavioral sciences was formed to lead this effort. An MDTT champion was identified and financially supported by the RCMI to lead this work. Our first step in identifying Morehouse SOM researchers who were working in an MDTT or MDTT-like team science structure was to provide information on MDTTs to our academic community. We believe semantics and a lack of understanding of the description of MDTTs prevents researchers from identifying that their work fits into this team science continuum. MDTT informational presentations were made in department meetings, workshops, and individual small groups. This informal information exchange provided feedback from researchers and staff, and we identified 16 teams that met our definition of a MDTT (See Table 1).

Next, each identified MDTT was assessed through an unstructured interview conducted by the MDTT champion. Researchers were asked to describe the composition of their teams, with specific attention paid to inclusion of student/leader members and types of community involvement. These unstructured interviews lasted from 15 min to one hour, depending upon the size of the team and the research topic.

We then enlisted the assistance of our Evaluation and Institutional Assessment Unit to further catalogue, categorize, and characterize the MDTTs at Morehouse SOM. Individual interviews were administered to the lead MDTT team member to assess the following characteristics of each MDTT: (a) project information (descriptions, titles), (b) departments participating in the MDTT, (c) members of the MDTT, (d) publications resulting from the MDTT, (e) role descriptions assigned to members of the MDTT, (f) accomplishments/successes of the MDTT, and (g) sustainability efforts.

The responses from these interviews were categorized and used to create a relational database to house current and future MDTTs via Microsoft Access. A relational database organizes data into rows and columns, which collectively form a table. Data is typically structured across multiple tables which can be joined together via a primary key or a foreign key. A primary key is a column in a table that uniquely identifies the rows of data in that respective table. A foreign key is a column in a table that refers to the primary key of another table. These unique identifiers demonstrate the different relationships which exist between tables. The characteristics of the MDTTs served as the tables that comprised the database. Each table and its data entries are connected through one-to-one relationships. In a one-to-one relationship, there is a connection between information in two tables, where each instance of data appears once. Each of the MDTTs have a designated MDTT ID (a primary key) to differentiate them, but also to create a relationship that would return related records when queried within the database. For example, the COVID—CEAL program has a primary key with a numerical code as its unique identifier. All data associated in other tables (roles, successes, sustainability efforts, etc.) share a relationship with this identifier. When prompted to provide information in other tables via filtering for this numerical identifier, search results will only provide data relating to the COVID—CEAL program. This relational database will allow us to track the progress of the MDTTs, including success measures and sustainability efforts. The next phase in this process is to follow up with each MDTT team to conduct a more detailed assessment and expand the MDTT database accordingly. A quantitative institution-wide survey will also be administered to identify additional MDTTs.

MDTT Examples at Morehouse SOM. As indicated previously, we have identified 16 MDTTs at Morehouse SOM. We will highlight four MDTTs that demonstrate the scope of team members, types of community involvement, and various impact outcomes below.

Case Study #1: Childhood Lead Exposure. During the RCMI 2022 Conference, we presented the research of a Morehouse SOM faculty member, Carmen M. Dickinson-Copeland, PhD, MSCR, “An MDTTs Approach to Addressing Low-Level Lead Exposure in Georgia Children”. Dr. Dickinson-Copeland’s research addresses the health inequities associated with lead exposure in children living in metro Atlanta neighborhoods. This MDTT is focused on the deleterious effects of low-level lead (LLL) exposure in children aged two to six years. Lead (Pb) is a pervasive environmental contaminant and potent neurotoxicant in children. Pb can impact child health across the range of possible exposures, and there is no safe level for children. This MDTT aims to identify predictors for sub-clinical blood lead levels (BLLs), between 2 and 5 micrograms/deciliter, as children with these BLLs are not currently considered “poisoned” but may still experience adverse effects.

This MDTT has four members, including a basic sciences researcher, clinician scientist, data scientist, and environmental scientist, across two academic institutions (Morehouse SOM and the University of Buffalo SUNY). The members were from the departments of Microbiology, Biochemistry, and Immunology, Pediatrics, and Medicine (Morehouse SOM), and Epidemiology and Environmental Health (University of Buffalo SUNY). This MDTT has worked with the RCMI Community Engagement Core to identify opportunities to foster community engagement and has participated in several community events to educate on the risks of childhood lead exposure. Additionally, there have been two undergraduate students who each spent a year working on this project. There have been some obstacles that the MDTT has had to navigate including geographic dispersion, bottlenecks related to task interdependence, and goal misalignment. Despite these obstacles, the MDTT has succeeded in publishing novel findings [9] and received additional funding for sustainability.

Case Study #2: Enhancing Student Learning Experiences One of the early challenges identified by Morehouse SOM was the difficulty in assuring student learners’ meaningful involvement in MDTT science research. The Quality Enhancement Plan (QEP) was part of the Morehouse SOM accreditation process and focused on Interprofessional Education Plan (IPE). Our Interprofessional Education, QEP is IPE is our approved Southern Association of Colleges and Schools Commission on Colleges (SACSCOC) Accredited (QEP). Real change and true sustainability do not occur within an institution without full acceptance and integration into the institution’s core mission. For health professions education, this change begins in their basic education curriculum. Competition with required curricula, lack of research exposure, faculty research focus, and time management for both students and faculty were all identified as hindrances to the increased involvement of student learners within MDTTs. We recognized that we not only wanted student MDTT involvement, but more importantly we wanted students to understand, participate in and build MDTTs throughout their future careers.

This in-process MDTT is led by our Office of Academic & Community Innovation (ACI) and includes members representing institutional administration, basic sciences and clinical departments, education, and student learners. Every entering student (MD, MPH, MS, PhD) is required to learn about and participate in an interprofessional educational experience during their first year of health professions training. Working in assigned learning communities, all first-year MD students are simultaneously required to complete a two-semester community health course at a community organization site. Through the Health Equity Activity Registry (HEAR), students are electively given the opportunity to join and interact with research teams throughout the institution. Introducing health professional students to the MDTT concept early in their academic journey will provide not only an immediate experience, but it will also have a perpetual impact throughout their professional careers.

Case Study #3: Natural Product Research Center. A unique Morehouse SOM MDTT is found within our Natural Products Research Center that conducts research with a team of African traditional healers throughout the continent of Africa through a collaboration with PROMETRA, an international non-governmental organization (NGO). PROMETRA is headquartered in Dakar, Senegal and is dedicated to preserving and restoring African traditional medicine and indigenous sciences. This MDTT brings important basic scientific knowledge to the villages of Africa and communities of America. Traditional healers provide approximately 80% of health care and health education to populations in Africa [10] and in that role are major providers and agents of health equity. Basic scientists, clinical practitioners, anthropologists, traditional healers, students, and patients living with HIV are all members of this bicontinental MDTT. Successes include a scientific publication, Identification of a Novel Anti-HIV-1 Protein from Morordica balsamina Leaf Extract in the International Journal of Environmental Research and Public Health [11], institutional and private funding exceeding $1M, intellectual property patent protection, and a major community education video documentary made in partnership with the Andrew J Young Foundation. The Morehouse SOM T^x^ research has identified a novel protein that is effective against HIV and other enveloped viruses [12]. This MDTT team science structure crosses oceans, languages, and cultures and presents challenges in terms of geography and communication. An MDTT that crosses such a wide swath of geography requires significant investments in sustainability.

Case Study #4: Fibroid MDTT: Laboratory to Patients to Community to Congress. Black women in the United States are up to four times more likely than white women to develop uterine fibroids. While noncancerous, fibroids can cause painful and excessive uterine bleeding, interfere with everyday life and self-image, and affect fertility. This MDTT contains members (faculty & learners) from two different medical schools, representing institutional administration, basic science and clinical departments. A major community advocacy group (The White Dress Project) and social scientists who advocate for budget and policies to increase the education and financial support for fibroid research are also part of this MDTT. The basic scientists conducted research to identify specific genes and the TGFβ pathway utilizing stem cell research in laboratories and the MDTT conducted community-based educational town halls. Later, MDTT members testified in the halls of the US Congress, our Morehouse SOM MDTT way of taking T^x^ research into the lives of the populations that we serve. Senator Elizabeth Warren and Congressman David Scott invited MDTT members to testify in a Congressional hearing, Women’s Health 201: The State of Gynecological Health in the US. A bill named the Uterine Fibroid Research and Education Act of 2020 was later introduced by then-Senator Kamala Harris, which awaits full passage by Congress. This MDTT has received multiple grants from government and private sources to continue its work and has successfully published its research.

## 3. Discussion

Utilizing U54 RCMI funding, Morehouse SOM made an intentional and concerted effort to identify, catalogue, and characterize MDTTs at the institution to further understand and document the ways Morehouse SOM was involved in leading the advancement and creation of health equity. As a result of this ongoing process, we have defined MDTTs at the institution, educated our community, and identified 16 existing MDTTs within the institution. Our long-term goal is to substantially increase the number of MDTTs. Through this process we have identified successes, challenges, and preliminary solutions.

Success Measurements A major success of MDTTs at Morehouse SOM has been the institutional buy-in for integration of student learners. This is evidenced by the intentional inclusion in the institution’s interprofessional education, and the inclusion in the accreditation process. Other successes include influence on policy that directly addresses health inequity with team members invited to present to the United States Congress, resulting in the introduction of legislation to address a significant health issue for women, particularly Black women. Success measures that we are currently monitoring for individual MDTTs include

Transition from an “in progress” MDTT to a full MDTT which includes student learners and community membersPeer reviewed and lay publicationsFunding received from all sourcesDegree of institutionalizationNumber of protocols, policies, and advocacy campaigns informed by the researchLevel of community involvementIntellectual property development

Challenges. This process of documenting MDTTs and engaging faculty members in discussions about MDTTs has revealed institutional challenges, not uncommon at other institutions engaged in team science. Institutional and academic behavior change is difficult to orchestrate. In the process of building and maintaining MDTTs throughout the institution, we identified teachable moments and challenges that hindered our MDTT team-building efforts through interviews and observations:The pressure for faculty to publish, obtain research outcomes, secure promotion, and seek funding consumed a great deal of researchers’ time, limiting embracing new ideas or adding additional student mentoring or community participation responsibilities.It is difficult to recruit student learner members for MDTT team participation. We identified this lack of student involvement and attributed it not to a lack of student willingness, but to an actual lack of curriculum time designated for non-required courses.The need to continually identify funding (especially in an under-resourced minority institution) is a priority task for researchers. Grant writing and fund development are time consuming activities. Current funding mechanisms of most government grants are not conducive to distributing funds across departments, providing community partnership grants, or providing unrestricted funds that can be used for innovation.Even with Morehouse SOM’s long history of community involvement, meaningful involvement of community members in our MDTT teams remains challenging. These partnerships require extensive time to assure that the research is clearly translated in addition to supporting the community partners to widely disseminate through noncomplicated funding mechanisms and support of organizational capacity development. Community members must be fully involved and fully respected as MDTT members.

Preliminary solutions to these challenges include institutionalizing MDTT concepts such as IPE into the required curricula, allowing students to participate in research groups; providing small incentive “jump start” grants to forming MDTTs; providing T^x^ pilot grant funding; and supporting our Morehouse SOM Mentoring Academy for faculty.

Sustainability of the initiative is key to advancing the progress of the identified MDTTs, identifying additional MDTTs, and promoting the creation of additional MDTTs. Institutional leadership has recognized the importance of MDTTs and has sought to secure funding to support the process of identifying and promoting MDTTs at Morehouse SOM. One of our next steps is to develop criteria for calculating the comprehensive return on investment (ROI) for individual MDTT activities.

## 4. Conclusions

MDTTs at Morehouse SOM help to address complex scientific problems, advancing science along the T^x^ continuum, hopefully leading to health equity. It is important to catalogue and characterize MDTTs at Morehouse SOM to identify barriers and facilitators to success, to facilitate implementation of strategies to maximize the success of MDTTs, and to promote the development of MDTTs. Study of social determinates of health clearly demonstrate that health conditions are not merely borne and exacerbated by biological factors. Quite the opposite, we know that health conditions are severely affected by environmental and political factors as well, as demonstrated most recently by the COVID-19 pandemic. To effectively address the complexity of health and healthcare, it is imperative that we not work in silos. Teams composed of various scientific disciplines, student learners, and community members will prove to be beneficial to advancing health equity. We see equal importance in studying the development of these MDTTs, understanding how MDTTs operate to maximize efficiency and impact, and proactively developing new teams in other areas of health disparities. We believe that utilizing this MDTT model allows us to increase the rapid translation of research discoveries to improve the health of minority, underserved populations, and disadvantaged communities.

## Figures and Tables

**Table 1 ijerph-20-04302-t001:** Morehouse SOM MDTT Database Projects List.

MDTT	Description Project/Purpose	Member Types *	Success to Date	Sustainability Efforts
COVID MDTTs				
National COVID-19 Resiliency Network (NCRN)	To collaborate with community-based organizations across the nation to mitigate the impact of the COVID-19 pandemic	AD, BehS, BS, C, COM, ED, PH, SS	44 community partners https://ncrn.msm.edu/ (accessed on 30 December 2022) National Advisory Board National Community Coalition Board Dissemination of culturally and linguistically appropriate information	Leveraged network to secure additional funding
COVID—CEAL	To understand factors that contribute to the disproportionate burden of COVID-19 in underserved communities and establish effective, community-engaged research and outreach response strategies.	AD, BehS, BS, C, COM, ED, PH, SS	Community coalition governing board 313 discrete national media presentations (TV, radio, social media, webinars, newspapers) 1700 community residents vaccinated and 225 enrolled in COVID Novovax vaccine trial	Initial funding $1 M/year Secured $1.4 M in additional funding
Natural Products Research Center (NPRC) MDTTs				
Medical Cannabis & Endocannabinoid Workgroup	To demonstrate Morehouse SOM’s multidisciplinary T^x^ approach to the understanding and study of the use of medical cannabis and the endocannabinoid system in the 65+ population.	AD, BS, C, COM, ED, LIB, PH, S/L, SS	Cannabinoid & Endocannabinoid Research Spectrum National Roundtable and Town Hall Feb 2021 Recruitment of a faculty endocannabinoid specialist (in process) Research partner with state selected medical cannabis producer	$2.75 M Kessler Research Foundation grant Pending funding from medical cannabis production companies
HIV & Emerging Pathogens	To expand our Morehouse SOM research in the areas of emerging pathogens, including HIV, Ebola, Zika, Malaria, Herpes and Coronavirus	AD, BS, C, IP, PH, S/L Global members	Identification, isolation & sequencing of 30 kD protein, MoMo30 COVID patent issued Macaque study completed Publication: https://doi.org/10.3390/ijerph192215227	Morehouse SOM Innovation Funds $986,086 over 3 years COVID supplement funded
RCMI Project MDTTs				
Reduced DNA Repair Capacity Leads to Increased Risk of Uterine Fibroids in African Americans	To identify genomic and epigenomic risk profiles associated with increased susceptibility to UFs in AA women and will inform an efficient precision medicine approach for development of novel preventative and therapeutic strategies.	AD, BS, C, COM, S/L	National Uterine Fibroid Town Hall Webinar https://www.youtube.com/watch?v=8jV7ntbbKs8US (accessed on 30 December 2022) Congressional Testimony 2019 1 Publication: Elkafas, Ali, Elmorsy et al. (2020) https://doi.org/10.3390/cells9061459	Submission of a Collaborative R01 to NIEHS Plans to submit a Multi-PI R01 to ICER
Project GRIT- Building Biobehavioral Goal Directed Resilience Among African American (AA) Women	To implement an innovative intervention to help expand existing paradigms to promote culturally centered, trauma-informed mental health care and wellness for AA women with PTSD.	BehS, C, COM, ED, PH, S/L, SS	4 scientific presentations made Established and disseminated Community Mental Health Equity Newsletter (Institutional Cross Disciplinary Collaboration Resilience: Black Women and Public Health (in press)	R01 submitted but not funded Ongoing development TA consultation with Morehouse SOM Office of Research Development
African American Women Treated with tPA	To investigate the effectiveness of TPA in African American women through a prospective study and to investigate blood transcriptome profiles to predict response to TPA in women	BS, C	The COVID pandemic halted patient identification and selection The team will supplement prospective study with retrospective analysis of tPA in African American women	NIH RO1 application submitted
Advancing Rural Health Disparities Through Innovation	To expand a multi-sectorial coalition to guide, advise and advance health disparities research in rural counties in southwest Georgia	AD, C, COM, ED, PH, BUS	Conducting community listening tours with key information interviews utilizing PhenXToolkit Conducted virtual Town Halls Women’s Health Fair with local hospital Use of Morehouse SOM’s Health 360x technology	Leveraging coalition for next round of RCMI funding
Childhood Lead Exposure An MDTTs Approach to Addressing Low-Level Lead Exposure in Georgia Children	Decrease health disparities associated with sub-clinical lead exposure through the identification of socioeconomic risk factors and molecular biomarkers in children	BS, C, PH, S/L AD, COM	1 Publication: Dickinson-Copeland CM, Immergluck LC, Britez M, Yan F, Geng R, Edelson M, Kendrick-Allwood SR, Kordas K https://doi.org/10.3390/ijerph18105163	Received T^x^ Pilot Project Award Pediatric and Reproductive Environmental Health Scholars-Southeastern Environmental Exposures and Disparities (PREHS-SEED K12)—$200,000
Other Morehouse SOM MDTTS				
Interprofessional Education QEP is IPE	To develop IPE experiences to prepare Morehouse SOM students for future health teams	AD, BS, C, ED, S/L	Included in Morehouse SOM ten-year accreditation plan. Required curricula component for all entering Morehouse SOM students	Institutional Support $1.304 million over 5 years
Office of Global Health Equity (OGHE)	To develop global learning experiences for Morehouse SOM students, faculty & staff to work with immigrant populations within the State of Georgia	AD, BehS, BS, C, COM, ED, IP, PH, S/L	Global Health Webinar Series—Five webinars conducted to date related to COVID-19	575,000 in institutional and extramural funding received
HBBB—HIV from Bench to Bedside and Beyond	To coordinate HIV activities across Morehouse SOM	AD, BehS, BS, C, COM, ED, PH, S/L	Collaboration with Emory University Center for AIDS Research (CFAR) on new Scientific Working Group—Health Equity Morehouse SOM MD4 Human Sexuality elective created Developing stigma reduction project in Black gay men using at-home anal swab testing	Leveraged COVID-19 Vaccine Trial infrastructure to launch HIV vaccine trial. infrastructure to launch HIV vaccine trial
Choice Neighborhood	To improve the economic, education and health outcomes of community residents.	AD, C, COM, PH, S/L	Community Health Worker teams for insurance navigation Involvement (patient referral) to Morehouse SOM student run HEAL Clinic (free clinic services) Presentation at Beyond Flexner Conference 2021	HUD continuation application submitted
Center for Maternal Health Equity at Morehouse SOM	To pursue equity in maternal health by reducing maternal morbidity and mortality locally, nationally and globally by building and strengthening community-academic partnerships, developing a competitive translational research program and offering interdisciplinary and professional training	BS, C, COM, ED	Development of a maternal mortality prevention and optimal reproductive health promotion mobile app—Prevent Maternal Morbidity & Mortality Mobile Technology (PM3) Creation of a simulation model—Respectful Maternity Care Developed Community-Based Perinatal Patient Navigator Training	Successful advocacy to obtain State of Georgia matching funds in the $500,000/year for FY 20 & FY21 and now in state base budget Institutional support of $500,000/year Extramural support from Johnson & Johnson, Deloitte, Amerigroup, Health Care Georgia and Goldman Sachs
Health Equity—T^x^ Work Group	To lead the work of Morehouse SOM’s T^x^ approach and philosophy to health equity	AD, BehS, BS, C, ED, PH,	Coordination of 2 successful rounds of T^x^ pilot projects to advance health equity in communities. Development of a national Health Equity Survey	Morehouse SOM institutional provided for pilot grants $75,000 per grant—7 funded to date = $525,000
Data Science Across the Atlanta University Center (AUC)	To advance data science on topics that impact Black American and to diversify the data science workforce	AD, BS, C, S/L	Seminar series focused on research, education and community engagement New course for AUC students—Data & the African Diaspora Development of Learning & Research Communities (LRC)	United Health Group AUC (HBCU) Data Partnership $8.25 M over five years

* Member Types: AD—administrator/staff, BehS—behavioral scientist, BS—basic science, BUS—business, C—clinical, COM—community, ED—education, IP—international partner, LIB—librarian, PH—public health, S/L—student/learner, SS—social scientist.

## Data Availability

Data is contained within the article.

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
