# Peer review of "Moving Morehouse School of Medicine Translation Tx Research through MDTTs—Multidisciplinary Translational Teams"

_ijerph, 2023, doi:10.3390/ijerph20054302_

Round 1

Reviewer 1 Report

Thank you very much for the opportunity to edit this wonderful article and addition to the science. I have a few suggestions from my reading below. 

Line 40, I cannot seem to find a definition for MSM, I assume it means Morehouse School of Medicine.

Line 53, has this work been done before? By whom? Is this a novel approach?

Line 72, How was this champion funded? Is this sustainable?

Line 109 Are there any citations for this?

Line 130 is there an example of how an MDTT conducted this work?

Line 177 I cannot seem to find a definition of the acronym QEP and IPE.  IPE was later identified in line 270.

Line 181 How was this information gained?

Line 191 I cannot seem to find a definition for GEBS?

I would, in general, ensure all acronyms are defined. See Line 203 (NGO for example).

Author Response

I cannot seem to find a definition for MSM, I assume it means Morehouse School of Medicine. We have updated the acronym to be consistent throughout the manuscript.

Line 53, has this work been done before? By whom? Is this a novel approach? We have clarified that the definition is Morehouse School of Medicine’s definition and what differentiates the definition from other team science definitions.

Line 72, How was this champion funded? Is this sustainable? We have added information that the champion was funded by RCMI grant funds. Additional information has been added in the discussion section to address sustainability.

Line 109 Are there any citations for this? No, this survey was developed by MSM.

Line 130 is there an example of how an MDTT conducted this work? Yes, we have provided four examples and revised the narrative to better describe the functioning of the MDTTs.

Line 181 How was this information gained? In point 2 – Methods. Please, better complement the description. For example, describe more details of how each of the issues was addressed. To those who were interviewed, with some instrument, how was the information analyzed?, etc). The methods section has been revised to clarify data collection methods.

I would, in general, ensure all acronyms are defined. See Line 203 (NGO for example).  All acronyms have been spelled out on first use.

Reviewer 2 Report

Many thanks to the authors for submitting the manuscript. The article provides a very relevant and necessary topic for the development of research considering a more comprehensive and sustainable approach.

I consider that the description of the manuscript could go much deeper into the work and experience of the research teams so that they can respond to the proposed objectives.

Here are some observations:

Both in the abstract, in the introduction, and throughout the document, it is recommended to review the use of abbreviations, since several (for example, MSM, RCMI, QEP, IP) are not easy to deduce or understand.

In point 2 - Methods. Please, better complement the description. For example, describe more details of how each of the issues was addressed. to those who were interviewed, with some instrument, how was the information analyzed?, etc)

Table 1 could be summarized a little better to make reading easier and the presentation more uniform. It could also be complemented, for example in members, mention the number of members, work dynamics, working time, etc.

In the results (lines 107-126) the authors wrote in the future about surveys that will be carried out. To make it easier to read, could you possibly move them up for discussion? in order to show the next steps?, and synthesize it a little better.

When mentioning the examples of the MDTT work groups, I think it would be more useful to describe the aspects related to the work experience of the group, rather than the scientific theme, and focus on the description that was mentioned in the methodology.
Likewise, in each group, some aspects could be described as challenges. currently, in the discussion, several points of challenges are raised, but they have not been previously presented.
Also in the discussion, it would be good to discuss the results with other contexts and previous experiences.

Author Response

Both in the abstract, in the introduction, and throughout the document, it is recommended to review the use of abbreviations, since several (for example, MSM, RCMI, QEP, IP) are not easy to deduce or understand.  All acronyms have been spelled out on first use.

In point 2 – Methods. Please, better complement the description. For example, describe more details of how each of the issues was addressed. To those who were interviewed, with some instrument, how was the information analyzed?, etc). The methods section has been revised to clarify data collection methods.

Table 1 could be summarized a little better to make reading easier and the presentation more uniform. It could also be complemented, for example in members, mention the number of members, work dynamics, working time, etc. Significant edits have been made to the table to streamline and make the information more uniformed.

In the results (lines 107-126) the authors wrote in the future about surveys that will be carried out. To make it easier to read, could you possibly move them up for discussion? In order to show the next steps?, and synthesize it a little better. This information has been moved and synthesized.

When mentioning the examples of the MDTT work groups, I think it would be more useful to describe the aspects related to the work experience of the group, rather than the scientific theme and focus on the description that was mentioned in the methodology. Likewise, in each group, some aspects could be described as challenges. Currently, in the discussion, several points of challenges are raised, but they have not been previously presented. Also in the discussion, it would be good to discuss the results with other contexts and previous experiences. We have revised the descriptions of the MDTTs to better address the work experience of the group. We have added some information on challenges for the MDTTs in the descriptions. We also decided to keep the challenges in discussion section. We believe this better helps to synthesize the information.

Reviewer 3 Report

This is a reflection/commentary to increase the rapid translation of research discoveries to improve health. As the objective is to disseminate and show the benefits of this reality, the document seems explicit to me.

In the authors' excerpt, it is clear that an INFORMAL identification allowed them to identify these teams. The request for clarification goes in this direction so that the authors can explain how this identification occurred—using some databases where these teams were identified? Teams working on these themes were recognised at congresses/conferences? More information was requested on the websites? How were these 16 identified? How can authors ensure that there are 16 and not 20, for example? That is, what was the process used to arrive at these 16?

Author Response

In the authors' excerpt, it is clear that an INFORMAL identification allowed them to identify these teams. The request for clarification goes in this direction so that the authors can explain how this identification occurred—using some databases where these teams were identified? Teams working on these themes were recognised at congresses/conferences? More information was requested on the websites? How were these 16 identified? How can authors ensure that there are 16 and not 20, for example? That is, what was the process used to arrive at these 16? The methods section has been revised to better explain how the 16 MDTTs were identified.

Reviewer 4 Report

This paper is a case study, not a commentary. Can it be re-classified as appropriate before it is sent for peer review?

Major comments:

P.2, line 53: What is the source of definition of MDTT? Please provide references.

Minor comments:

Please proofread the whole manuscript so that it is ready for submission and peer review. Examples of problematic sentences:

-          P. 2, lines 47: “.. define and constricted…’’

-          P.2, line 65: “… in increasing the journal from…”

Author Response

2, line 53: What is the source of definition of MDTT? Please provide references. We have clarified that the definition is Morehouse School of Medicine’s definition and what differentiates the definition from other team science definitions.

Please proofread the whole manuscript so that it is ready for submission and peer review. Examples of problematic sentences:

-          P. 2, lines 47: “.. define and constricted…’’

   P.2, line 65: “… in increasing the journal from…”

The manuscript has been proofread.

Round 2

Reviewer 2 Report

Many thanks to the authors for incorporating the suggestions made previously.
I have no further comments. It would be important to review the wording since when editing the text it seems to me that some things have been inconsistent.
On the other hand, it is important to check if it falls under the category of "commentary", or rather a "case report".

Author Response

We have made significant revisions to the manuscript and the table. The manuscript is being submitted as a case study.

Reviewer 4 Report

As commented previously, this paper is not a commentary. Please recategorize it as appropriate before sending it out for review.

A lot of texts in Table 1 are cut off and cannot be read.  

There are many grammatical and editing errors in this revision. The whole manuscript requires proper proofreading before it should be resubmitted for review.

Author Response

The paper has been undergone substantial revisions and is being submitted as a case study.

The table has been reformatted

The paper has undergone several revisions and reviews for grammar and spelling.